# Zero-Shot Character Identification and Speaker Prediction in Comics via Iterative Multimodal Fusion

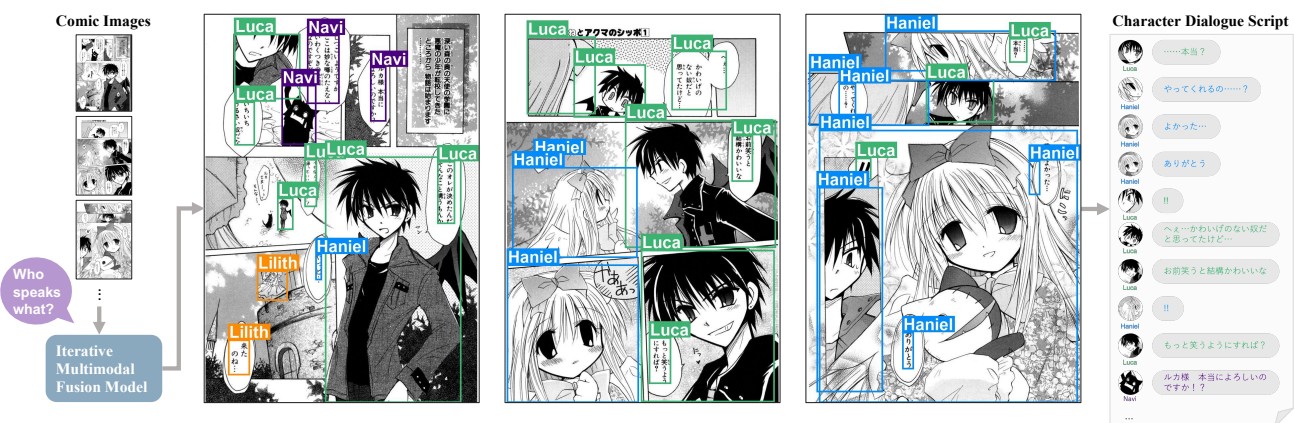

**Figure 1: Our framework can predict character labels of unseen comics only from images. Courtesy of Kiriga Yuki.**

## ABSTRACT

Recognizing characters and predicting speakers of dialogue are critical for comic processing tasks, such as voice generation or translation. However, because characters vary by comic title, supervised learning approaches like training character classifiers which require specific annotations for each comic title are infeasible. This motivates us to propose a novel zero-shot approach, allowing machines to identify characters and predict speaker names based solely on unannotated comic images. In spite of their importance in real-world applications, these task have largely remained unexplored due to challenges in story comprehension and multimodal integration. Recent large language models (LLMs) have shown great capability for text understanding and reasoning, while their application to multimodal content analysis is still an open problem. To address this problem, we propose an iterative multimodal framework, the first to employ multimodal information for both character identification and speaker prediction tasks. Our experiments demonstrate the effectiveness of the proposed framework, establishing a robust baseline for these tasks. Furthermore, since our method requires no training data or annotations, it can be used as-is on any comic series.

## CCS CONCEPTS

• **Applied computing → Media arts**.

## KEYWORDS

Comics understanding, Multimodal content analysis, Zero-shot learning, Speaker prediction, Character identification

## 1 INTRODUCTION

The global comics market has experienced significant growth, sparking interest in computational analysis of comics to enrich user experience and accessibility. Given that characters are central to comic storytelling, this paper focuses on two key tasks: *character identification*, recognizing the characters in images, and *speaker prediction*, predicting the speakers of specific dialogue. These tasks enable diverse applications like voicing comics with character-specific voices and machine translations capturing each character's unique speech style.

Previous studies on speaker prediction in comics primarily addressed the correspondence between character regions and speech bubble regions in images [8, 17], but they did not determine the speakers' names. If we want to know the character names of the speakers, we first need to identify the character names of the character regions. The straightforward supervised approach involves training models to detect and classify characters in images [13, 25]. However, this approach requires annotations for each comic with different characters, which is impractical in the fast-paced comic industry that releases thousands of titles annually.

This motivates us to tackle the problem as a *zero-shot* task: *identifying characters and predicting speakers in new, unseen comics simply by analyzing the images, without need for any prior annotations*.

How does a human recognize speakers in new comics, without prior knowledge? The process typically starts with noticing character names in the dialogue. For instance, if one character calls another "Naruto", we associate the response to that dialogue with Naruto. We then connect the character's visual appearance with

their dialogue. When Naruto reappears in the following pages, even if his name is not mentioned in the dialogue, we can still know the character named Naruto. Context from dialogues is also an important clue. Once we have determined the speaker of a particular dialogue, we can use contextual information to predict the speakers of nearby dialogues. Additionally, this process allows us to update our knowledge of the characters' visual appearances across different pages. After seeing many examples, we learn to identify characters and predict dialogue speakers even when their names are not directly mentioned in the text.

This process suggests two key challenges. (1) *High-level text understanding*: To predict speakers from limited cues (e.g., names mentioned in dialogue), the system must interpret complex character interactions and story context throughout the whole book or chapter. (2) *Multimodal integration*: To identify characters and predict the name of speakers without any annotations, integrating both visual and textual information is essential, thereby making a combination of text and image modalities indispensable.

Addressing these challenges, we propose a multimodal fusion approach. We leverage large language models (LLMs) for their context understanding and reasoning capabilities [6, 20, 23], providing a strong baseline for zero-shot speaker prediction. To address the challenges of integrating LLMs with other modules and enhance the machine's comprehension of comics, we introduce an iterative framework. We merge text-based LLM predictions with image-based classifiers, and alternately refine each module using results from the other. This multimodal integration not only enables zero-shot character identification but also also notably improves text-only baseline in speaker prediction. Moreover, by iteratively refining the integration of text and image information, this approach enhances the utilization of both modalities, thereby deepening the machine's comprehension of comics.

Our contribution is twofold. (1) **New tasks:** We are the first to integrate the tasks of character identification and speaker prediction in comics. Furthermore, our approach tackles zero-shot tasks without requiring any training or annotations, which are directly applicable to real-world scenarios. (2) **Iterative multimodal fusion:** We pioneer in revealing the potential of LLMs for comics analysis and propose a novel method that integrates text and image modalities. To enhance the machine's deep understanding of comic content, we introduce an iterative framework aimed at progressively refining performance. This is the first study to use both text and image information for character identification and speaker prediction, which are unexplored even outside zero-shot settings.

## 2 RELATED WORK

Given the novelty of our study in predicting speaker names in comics, there are no existing studies directly related to this task. Therefore, we first explore related works from two distinct perspectives: comic speaker prediction and comic character identification. Then, we introduce works related to our proposed method.

### 2.1 Comic Speaker Prediction

Previous studies of comic speaker prediction focus on predicting the correspondence between character bounding boxes and text regions. Conventional methods are rule-based, primarily relying on the distance between the character and the text region [17]. They disregard visual and textual context information, causing them to fail in cases where the speaker is not closest to the text region. Recently, Li et al. employ scene graph generation (SGG) models [8] known for their effectiveness in visual relationship detection tasks [7, 18, 21, 22], to predict the correspondence in a more robust way by using visual semantic information such as the face angle. While the previous studies focused only on predicting the correspondence between character bounding boxes and text regions, we propose a new task to predict speaker identities, i.e., character names. This allows more applications such as text-to-speech with character-specific voices. Furthermore, while previous studies have utilized only visual information, we are the first to utilize both visual and textual information.

### 2.2 Comic Character Identification

Comic character identification is more challenging than face identification in real-world images due to the variance of drawing styles and poses. Zheng et al. constructed a cartoon-face dataset and proposed an image classification model to classify characters [25]. However, their supervised approach requires training character classifiers for each comic title, which limits the feasibility of this approach. Some previous studies attempt unsupervised methods to handle unseen comics. Tsubota et al. used deep features trained for generic face recognition in comics and adapted them to unseen comics [19]. Zhang et al. improved clustering performance on comics with a face-body combination and spatial-temporal correction [24]. However, they are limited to grouping the characters by clustering and cannot identify character names. In addition, even though dialogue is an important cue to identify characters in comics, existing approaches have not used textual information. In this paper, we propose the first multimodal approach that utilizes dialogue and predicts character names in unseen comics.

### 2.3 Large Language Models

The recent success of ChatGPT [14] and GPT-4 [15] has demonstrated the power of LLMs in understanding, generating, and interpreting human language with remarkable accuracy. Inspired by these advancements, we have pioneered the application of LLMs to the dialogue analysis in comics. Alongside the development of LLMs, large multimodal models (LMMs) such as LLaVA [9, 10] and MiniGPT-4 [26] have garnered attention for processing multimodal information. In spite of their capabilities in visual understanding and reasoning, currently, they can only handle a single or a few images as input. Comic analysis requires comprehension across longer contexts, such as multiple pages. Also, it involves learning character identity throughout the book or series. We thus propose an iterative framework to integrate multimodality and longer contexts into LLMs inference for this task.

## 3 APPROACH

### 3.1 Problem Settings

Let us define our problem setting for zero-shot character identification and speaker prediction. The inputs are a sequence of page

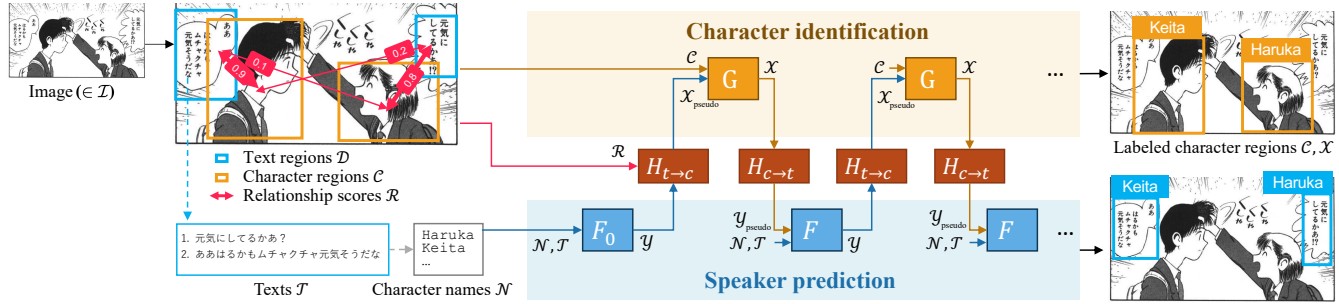

**Figure 2: Our proposed framework for zero-shot character identification and speaker prediction in comics. Courtesy of Ito Shinpei.**

images of a specific comic $\mathcal{I}$. The output consists of character regions $C = \{c_i\}_{i=1}^{N}$, text regions $\mathcal{D} = \{d_j\}_{j=1}^{M}$, and corresponding character labels $\mathcal{X} = \{x_i \in \mathcal{N}\}_{i=1}^{N}$ and $\mathcal{Y} = \{y_j \in \mathcal{N}\}_{j=1}^{M}$. The character regions are the bodies of characters, and the text regions are speech bubbles. $\mathcal{N}$ is the list of character names (e.g., $\mathcal{N} = \{\texttt{Keitaro}, \texttt{Naru}, \ldots\}$), which is extracted from dialogues. These names serve as target labels for speaker prediction and character identification tasks in the following steps. $N$ and $M$ denote the number of character and text regions in a given comic.

***Data preprocessing.*** Before initiating our main pipeline, we conducted a series of preprocessing steps. First, character regions $C$ and text regions $\mathcal{D}$ are obtained using object detectors for general comic character and text category, which are shown to achieve high accuracy [13]. Subsequently, we derived the initial relationship scores $\mathcal{R}$ from $\mathcal{I}$, $C$, and $\mathcal{D}$ using the SGG models [8]. Relationship scores $\mathcal{R} = \{r_{i \to j}\}_{i,j}$ represent the correspondences between character and text regions, where $r_{i \to j} \in (0,1)$ is the confidence that the character $c_i$ is the speaker of the text $d_j$. This is used to propagate labels between the character and text regions. Additionally, we extracted texts $\mathcal{T} = \{t_j\}_{j=1}^{M}$ from text regions $\mathcal{D}$ utilizing optical character recognition (OCR). Through text analysis by LLMs, we obtained the list of character names $\mathcal{N}$ from $\mathcal{T}$.

## 3.2 Overall Framework

Our framework for zero-shot speaker prediction and character identification is illustrated in Figure 2. Speaker prediction and character identification are performed iteratively. By using the output of each task as input for the other, we can exploit complementary multimodal information in both tasks, which leads to good performance even in zero-shot settings.

Our framework is comprised of three modules:

- **Speaker prediction:** Predict labels of text regions $\mathcal{Y}$ with LLMs. Initial predictions only use textual content, denoted as $F_0 : (\mathcal{T}, \mathcal{N}) \mapsto \mathcal{Y}$. From the second prediction, labels obtained from character identification $\mathcal{Y}_{\text{pseudo}}$ is used, denoted as $F : (\mathcal{T}, \mathcal{N}, \mathcal{Y}_{\text{pseudo}}) \mapsto \mathcal{Y}$.
- **Character identification:** Predict labels of character regions $\mathcal{X}$ using image information and pseudo labels $\mathcal{X}_{\text{pseudo}}$ obtained from speaker prediction. Denoted as $G : (C, \mathcal{X}_{\text{pseudo}}) \mapsto \mathcal{X}$.

---

**Algorithm 1** Overall framework

**Input**: Images $\mathcal{I}$
**Parameter**: Iteration times $n$
**Intermediate Output**: Texts $\mathcal{T}$, Character names $\mathcal{N}$, Relationship scores $\mathcal{R}$
**Output**: Character regions $C$, Text regions $\mathcal{D}$, Character labels $\mathcal{X}$, Text labels $\mathcal{Y}$

1: Data preprocessing
   $C, \mathcal{D} \leftarrow$ Object detection on $\mathcal{I}$
   $\mathcal{R} \leftarrow$ Initial relationship detection from $\mathcal{I}, C, \mathcal{D}$
   $\mathcal{T} \leftarrow$ OCR extraction from $\mathcal{I}, \mathcal{D}$
   $\mathcal{N} \leftarrow$ Character name extraction from $\mathcal{T}$
2: Initial speaker prediction: Get initial labels of $\mathcal{Y}$
   $\mathcal{Y} \leftarrow F_0(\mathcal{T}, \mathcal{N})$
3: **while** Iteration times $\leq n$ **do**
4:    Multimodal character identification: Update $\mathcal{X}$
      $\mathcal{X}_{\text{pseudo}} \leftarrow H_{\text{t}\to\text{c}}(\mathcal{R}, \mathcal{Y})$
      $\mathcal{X} \leftarrow G(C, \mathcal{X}_{\text{pseudo}})$
      $\mathcal{R} \leftarrow f_{\text{rescore}}(\mathcal{R}, \mathcal{X}, \mathcal{Y})$
5:    Multimodal speaker prediction: Update $\mathcal{Y}$
      $\mathcal{Y}_{\text{pseudo}} \leftarrow H_{\text{c}\to\text{t}}(\mathcal{R}, \mathcal{X})$
      $\mathcal{Y} \leftarrow F(\mathcal{T}, \mathcal{N}, \mathcal{Y}_{\text{pseudo}})$
      $\mathcal{R} \leftarrow f_{\text{rescore}}(\mathcal{R}, \mathcal{X}, \mathcal{Y})$
6: **end while**
7: **return** $C, \mathcal{D}, \mathcal{X}, \mathcal{Y}$

---

- **Label propagation:** Convert labels between character and text regions using relationship scores. Denoted as $H_{\text{t}\to\text{c}} : (\mathcal{R}, \mathcal{Y}) \mapsto \mathcal{X}$ and $H_{\text{c}\to\text{t}} : (\mathcal{R}, \mathcal{X}) \mapsto \mathcal{Y}$.

The procedure of our framework is shown in Algorithm 1. First, we predict speakers only from text information. This output is converted into labels of character regions using label propagation, whereby character identification is performed. Then, we predict speakers again using the labels obtained in the previous step. In addition, relationship scores are updated with $f_{\text{rescore}}$ based on the predicted labels. These processes are repeated for a specific number of iterations. In the following sections, we explain the details of each step.

**Input**

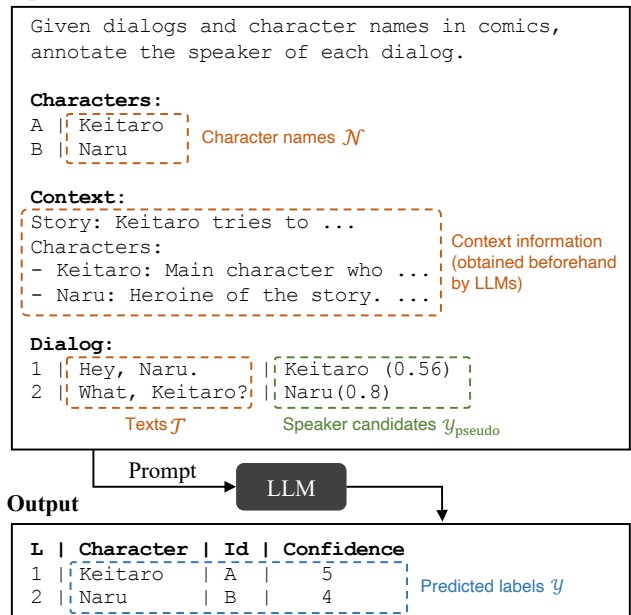

**Output**

Figure 3: Speaker prediction with LLMs.

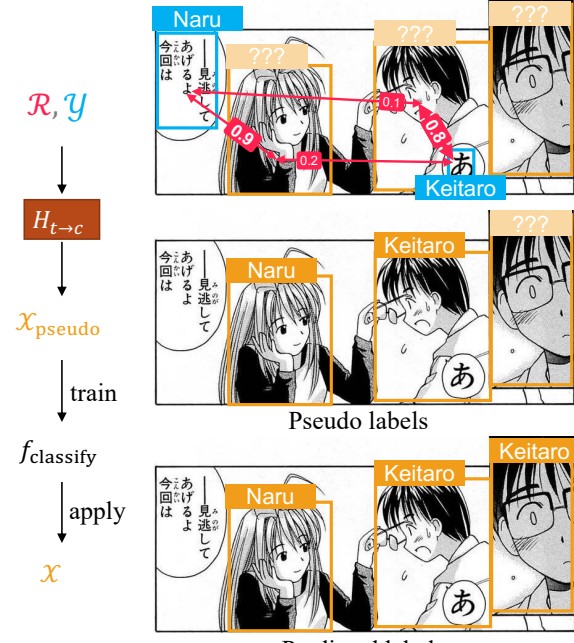

Figure 4: Pipeline of the character identification. Courtesy of Akamatsu Ken.

### 3.3 Initial Speaker Prediction

We first predict speakers using only the textual content using LLMs as $F_0(\mathcal{T}, \mathcal{N})$. GPT-4 [15] is used in this work. The input and outputs of LLMs are shown in Figure 3. We input the dialogues $\mathcal{T}$ and the list of appearing characters' names $\mathcal{N}$ into the LLMs. Due to output text length limitations of GPT-4, we split the dialogues into chunks. To compensate for the missing context about the story in each chunk, we first extract context information about the story summary and character profile from $\mathcal{T}$ and $\mathcal{N}$ using LLMs. This context is then fed into LLMs with each chunk as shown in Figure 3. Note that speaker candidates $\mathcal{Y}_{\text{pseudo}}$ are not input in the initial prediction. As the output, we let LLMs output both character IDs and character names to get a stable output. These output character IDs are converted into $\mathcal{Y}$. In addition, we let LLMs output integer confidence scores from 1 to 5, and we exclude data with low confidence from subsequent steps.

### 3.4 Multimodal Character Identification

Given initial speaker prediction results $\mathcal{Y}$, we perform character identification as shown in Figure 4.

The first step is pseudo label generation with the label propagation module $H_{t \to c}$. For each character region, we select the character-text pair that has the highest relationship score among all combinations involving this character region. Then, we use the selected text region's label as the pseudo label of the character region. To get higher precision, we set a threshold for the confidence of text region's label obtained by LLMs. In our experiments, removing data with confidence less than 3 lead to higher precision without reducing recall.

In the second step, we construct a character classifier to identify the characters with unknown labels. We train an image classifier $f_{\text{classify}}$ with pseudo labels $\mathcal{X}_{\text{pseudo}}$ and then obtain $\mathcal{X}$ by applying $f_{\text{classify}}$ to character regions $C$. We use ResNet50 [4] as the image classifier.

As $\mathcal{Y}$ becomes closer to the true text labels, the generated pseudo labels will also be closer to the ground truth. Correspondingly, the training data for the character classifier becomes progressively more reliable, resulting in higher identification accuracy. We iteratively enhance the performance of character identification by improving the precision of the speaker labels predicted by the speaker prediction module.

### 3.5 Multimodal Speaker Prediction

In the multimodal speaker prediction, we first generate pseudo labels for text regions $\mathcal{Y}_{\text{pseudo}}$ from $\mathcal{X}$ by using the label propagation module $H_{c \to t}$ in the same step as $H_{t \to c}$. We treat these pseudo labels as candidate speakers for the texts, taking the confidence of character identification to be the confidence for the speaker candidates. Confidences with less than 0.5 are filtered out from candidates.

Given $\mathcal{Y}_{\text{pseudo}}$, the speaker is predicted by LLMs with the prompt in Figure 3. We feed the LLMs both the confidence of character classifiers and the character name in a format such as `Keitaro (0.56)`, which informs the LLMs about the reliability of the given speaker candidates.

## 3.6 Relationship Rescoring

Relationship scores $\mathcal{R}$ are updated based on the predicted labels $\mathcal{X}$ and $\mathcal{Y}$. Since character labels are not considered in the initial relationship scores, more accurate relationship scores can be obtained by considering character labels. We use a simple rescoring method: $r_{i,j}$ is multiplied by scale $s$ if $x_i = y_j$ and divided by $s$ if $x_i \neq y_j$. The scale $s$ is computed from confidences of labels $p_{x_i}$ and $p_{y_j}$ as $\min(1, \lambda p_{x_i} p_{y_j})$, where $\lambda$ is a hyper-parameter that is set to 2 in our experiments. With this step, pairs that are predicted as having the same character labels gets higher relationship scores, and vice versa. This is similar to the process where humans recognize characters and predict the speaker-text correspondences when reading comics.

## 4 EXPERIMENTS

### 4.1 Experimental Setup

**Dataset.** We used the Manga109 dataset [1, 13], which comprises 109 volumes of Japanese comics and provides character labels for both character and text regions. For the zero-shot setting, we constructed the training and testing sets from distinct comic titles; the characters in the testing set were unseen in the training set. We selected 23 volumes as the testing set. The remaining volumes were used for training and validation of the relationship detection module and pre-training of the character classifier.

For the character regions, we used the body region annotations [13], which represent the entire bodies of the characters. We preferred this method because whole-body regions offer more information (such as clothing and body types) for character identification than face regions. Besides, comic datasets often include characters with body-only annotations, such as those shown from the back or non-human characters, which necessitates the use of body regions for identification. For the character labels associated with the text regions, we used Manga109Dialog annotations [8].

**Task Settings.** For each comic in the test set of Manga109, we used page images $\mathcal{I}$ and a list of character names as inputs. Since object detection and OCR are not the main focuses of this paper, and since the results of character name extraction can significantly impact the accurate evaluation of our main pipeline's performance, in the main evaluation, we omit the steps of obtaining $C$, $\mathcal{D}$, $\mathcal{T}$, and $\mathcal{N}$ in data preprocessing. Instead, we regard them as known information by utilizing the annotations of Manga109. This simplifies the evaluation and makes the experiment replication easier while maintaining the main challenges of our tasks. End-to-end zero-shot settings were evaluated in the final experiments. Given that some titles in Manga109 feature a large number of characters, we excluded characters with an appearance frequency of less than 3% from the list of character names in our experiments. However, to ensure a fair comparison with future works, we still predicted and evaluated the labels of these infrequently appearing characters.

Under the above settings, the tasks of speaker prediction and character identification were classifying the character labels for the character and text regions ($\mathcal{X}$ and $\mathcal{Y}$). We calculated the accuracy of classification results to evaluate these tasks, which is the ratio of correctly predicted regions to total regions. Additionally, we calculated precision and recall for the pseudo labels in the ablation study. Precision is the ratio of correct pseudo labels to the total number of generated pseudo labels. Recall is the ratio of correct pseudo labels to the total number of regions.

**Relationship Prediction.** Our method used the relationship scores $\mathcal{R}$ to show correspondences between character and text regions. We compared three types of initial relationship scores: **SGG:** a deep learning-based relationship prediction using the scene graph generation (SGG) model [8], **Distance:** a rule-based method using the distance between the center coordinates of the character and text regions [17], and **GT:** ground truth annotations of Manga109Dialog. We used SGG in the main experiments and used Distance and GT in the ablation study. The confidence of SGG model was used as the score for SGG, and the reciprocal of the distance was used as the score for Distance. In the case of GT, we took the relationship score to be 1.0 for all pairs.

**Speaker Prediction with LLMs.** We used the GPT-4 model [15] (specifically, `gpt4-0314`) for speaker prediction. First, we produced a context summary by feeding all dialogue to GPT-4. Next, we performed speaker prediction by inputting both the extracted context information and the dialogue text itself into the model. Due to GPT-4's output token limitation, we divided every conversation into segments, each comprising 60 sentences. A complete list of these prompts is available in the supplementary material.

**Character Classifier Training.** We used ResNet50 [4] as the character region classifier, initially pre-trained on ImageNet [3]. We chose this model as the classifier due to its robustness in handling the diverse and complex visual patterns of comic character bodies, which provides more advantages over simply fine-tuning face recognition models. We also explored state-of-the-art classification models like ConvNeXt[11] and models pre-trained on the anime dataset Danbooru[2]. However, our experiments demonstrated that the fine-tuned ResNet model offers the best performance.

Our training process involved two steps: pre-training for generic comics and fine-tuning for individual unseen comic in the test set. Pre-training, done only once for generic comics, aims at domain adaptation from the ImageNet pre-trained model to comic characters. Manga109 training set with ground truth annotations of 349 characters is used for pre-training. We fine-tune on each individual unseen comic using pseudo labels generated through our multimodal iterative fusion process. We employ various techniques, including data augmentation and model ensembling, to achieve stable results from training with noisy labels. Since this module itself is not the main focus of this paper, we describe the details in the supplementary material.

**Baselines.** Since no existing method can predict character labels in a zero-shot setup, we constructed our own baselines. For character identification task, we first group character regions of each comic by the clustering of deep features obtained using the Manga109 pre-trained model explained above. K-means clustering [12] with $k=|\mathcal{N}|$ is used. Since it is impossible to assign character labels to each cluster without text information, we map each cluster to the character labels using ground truth so that accuracy is maximized, which can be regarded as an upper bound of the clustering approach. The labels of character regions are converted into those of text regions using relationship prediction methods [8, 17], which are referred

| | iter | text | img | Speaker pred. | | | Character id. | | |
|---|---|---|---|---|---|---|---|---|---|
| | | | | *Easy* | *Hard* | *Total* | *Easy* | *Hard* | *Total* |
| **Baseline** | | | | | | | | | |
| K-means+Distance | - | | ✓ | 34.5* | 31.8* | 33.1* | 37.0* | 36.7* | 36.8* |
| K-means+SGG | - | | ✓ | 36.7* | 34.8* | 35.7* | 37.0* | 36.7* | 36.8* |
| **Proposed** | | | | | | | | | |
| LLM only | 0 | ✓ | | 41.8 | 45.1 | 43.6 | - | - | - |
| Multimodal | 1 | ✓ | ✓ | 51.0 | 51.2 | 51.1 | 45.8 | 39.6 | 42.4 |
| | 2 | ✓ | ✓ | 52.4 | **51.3** | **51.8** | 48.5 | **40.3** | **44.0** |
| | 3 | ✓ | ✓ | **53.5** | 49.8 | 51.6 | **48.9** | 37.7 | 42.8 |

(a) Results on different test sets. * indicates that the baseline method used the ground truth to map clusters into labels, as explained in the experimental setup.

| | iter | Speaker pred. | Character id. |
|---|---|---|---|
| **Baseline** | | | |
| K-means+GT | - | 42.0* | 36.8* |
| **Proposed** | | | |
| LLM only | 0 | 43.6 | - |
| Multimodal | 1 | 60.2 | 53.9 |
| | 2 | 63.4 | 55.5 |
| | 3 | **63.8** | **56.6** |

(b) Results using the ground truth relationships.

**Table 1: Speaker prediction and character identification accuracy (%).**

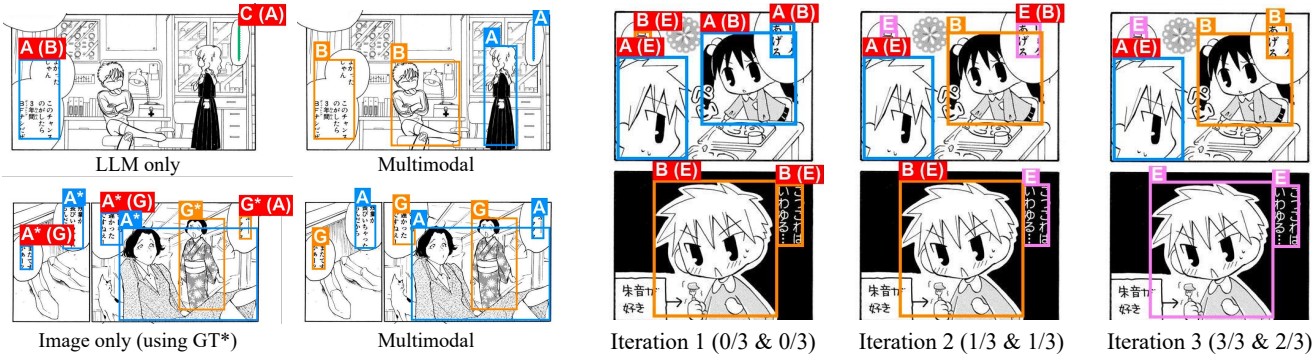

(a) Unimodal vs. Multimodal

(b) One-Step vs. Iterative (Accuracy of speaker pred. & character id.)

**Figure 5: Examples of prediction results. Courtesy of Tashiro Kimu, Hikochi Sakuya, Tenya. Color of the bounding box indicates the predicted character label. Red labels indicate failure predictions. The labels on the boxes (e.g., 'A') are character labels. Labels in brackets are the ground truth. (e.g., 'A (B)' is the case where the ground truth is B but the prediction is A.)**

to as **K-means + Distance** and **K-means + SGG**, respectively in Table 1.

## 4.2 Main Results

Table 1 shows the results of the proposed method and the baselines. The *text* and *img* columns indicate the used modalities for each method. Each iteration involves identifying characters based on the speaker predictions from the previous iteration, followed by a new round of speaker prediction that uses the updated character labels and relationship scores. The initial phase, where the speaker prediction is conducted only with textual information via LLMs, is iteration 0. The subsequent complete iteration cycle following iteration 0 is iteration 1. To validate the effectiveness of our proposed method, we divided the test set (*Total*) into *Easy* and *Hard* by the difficulty of relationship prediction. *Easy* contains 11 volumes with an accuracy of relationship prediction over 75%. The remaining 12 volumes were categorized as *Hard*.

As shown in Table 1 (a), our proposed multimodal approach produces a significant improvement in accuracy over unimodal methods in both speaker prediction and character identification.

For speaker prediction, the accuracy of *LLM only* was 43.6% but it increased to 51.1% in the first iteration. The accuracy of character identification also reached 42.4%, outperforming the baseline that uses the ground truth labels. Although the accuracy of character identification was less than 50%, it is noteworthy that our method is the first one that recognizes character labels in unseen comics. The difficulty of identifying less frequent characters in zero-shot settings causes the low accuracy in both tasks, which is posed as future work.

Across all test sets, the results from iteration 2 showed an improvement over iteration 1. In iteration 3, while the accuracy for *Easy* data continued to increase, there was a slight decline observed for *Hard* and *Total* data. This suggests that there are limits to the accuracy gains in relationship detection from further iterations To investigate this further, we do an analysis using the ground truth relationships in Table 1 (b), where the accuracy of both speaker prediction and character identification keeps increasing with further iterations. These results suggest our iterative process is more effective in the case that the prediction of relationships between text and character regions is accurate.

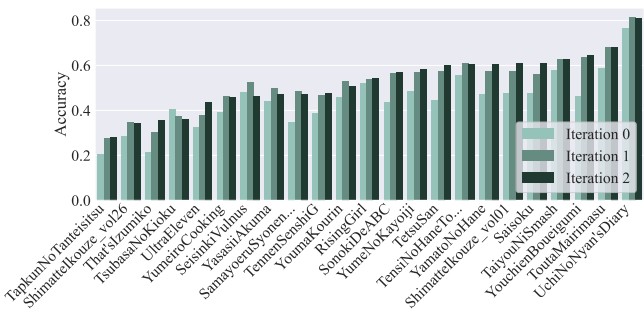

**Figure 6: Speaker prediction accuracy of each comic title.**

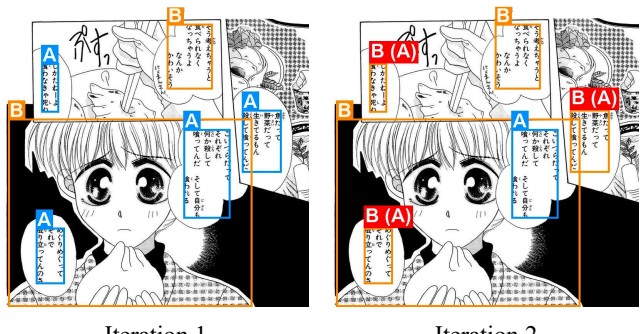

Iteration 1 Iteration 2

**Figure 7: Failure cases. Courtesy of Hanada Sakumi.**

Figure 5 shows examples of prediction results. We evaluate the effectiveness of the proposed method in two aspects: **Unimodal vs. Multimodal** (Figure 5 (a)) and **One-Step vs. Iterative** (Figure 5 (b)). Given that unimodal approaches are incapable of achieving zero-shot character identification, when using only LLM, character identification is not performed. *Image only* results corresponds to the baseline k-means+SGG: labels of character regions are derived from ground truth. Speaker predictions from texts alone present challenges for LLMs, especially with texts lacking distinctive character features, while it is easy to make correct predictions when integrating visual information. Conversely, when speaker predictions are made only based on images, such as by selecting the character closest to the text as the speaker, the method fails in two scenarios: (1) when the speaker is not the character closest to the text; (2) when the speaker does not appear in the image. In contrast, our multimodal approach handles these complexities well. Figure 5 (b) shows the results of the proposed method in each iteration. As the number of iterations increases, there is a clear trend of increasing accuracy in both character identification and speaker prediction tasks. Although the first iteration yields no correct predictions, by the third iteration, the accuracy has significantly improved. These findings affirm that our method can improve the accuracy of both tasks by iteratively refining the results using multimodal information.

Figure 6 shows the accuracy of speaker prediction for each comic. The three bars in each comic indicate the accuracy of iteration 0 (LLM only), iteration 1, and iteration 2, respectively. The iterations improved accuracy on most of the comics, which shows the effectiveness of our iterative approach on different comics. This figure

| | Speaker candidates | | | Speaker pred. |
|---|---|---|---|---|
| Relation | Prec | Recall | Accuracy | Accuracy |
| Distance | 60.7 | 29.0 | 44.6 | 48.5 |
| SGG | 60.8 | 32.4 | 45.1 | 51.1 |
| GT | 74.0 | 39.5 | 54.6 | 60.2 |

**Table 2: Precision and recall of speaker candidates and the speaker prediction accuracy examined by varying the relationship scores.**

| | Pseudo labels | | | Character id. |
|---|---|---|---|---|
| Relation | Prec | Recall | Accuracy | Accuracy |
| Distance | 32.6 | 29.1 | 29.9 | 41.6 |
| SGG | 34.2 | 28.4 | 29.2 | 42.4 |
| GT | 58.8 | 23.5 | 25.1 | 53.9 |

**Table 3: Precision and recall of pseudo labels and impact on character identification accuracy.**

also shows that performance can vary significantly across different comics, ranging from 0.2 to 0.8. Note that more iterations do not necessarily lead to higher accuracy. This is because if the accuracy of relationship prediction or character identification is low, the candidates can introduce noise into speaker prediction. An example is shown in Figure 7. In iteration 1, the LLMs correctly predicted the speaker (character A). However, due to the distance between the speaker and the text, character A received a low relationship score. In contrast, character B, being closer and having a higher relationship score, was incorrectly identified as the speaker candidate with high confidence. This led to an incorrect prediction in iteration 2, which explains why there is a decrease in accuracy by iterations in Figure 6 and Table 1.

## 4.3 Ablation Study

***Pseudo Labels.*** Table 2 and 3 shows the quality of the pseudo labels $X_{\text{pseudo}}$ and speaker candidates $\mathcal{Y}_{\text{pseudo}}$ and their effect on accuracy for speaker prediction and character identification tasks. Three relationship scores are compared. 2nd and 3rd columns are the precision and recall of the pseudo labels. The accuracies listed in the 4th column are those of the pseudo labels before thresholding. These are shown for comparison with the final performance score (5th column). The carryover from pseudo labels' accuracies (4th column) to final accuracy (5th column) explains the effectiveness of our iterative process (+6.0% and +13.2% gains in speaker prediction and character identification, respectively when using SGG). It shows that the result obtained from the previous step is refined in each module by using different modal information. By comparing relationship scores, we can see the result gets better by improving the quality of pseudo labels with better relationship prediction; specifically, the precision of GT in Table 3 is 24.6% higher than that of SGG, which leads to 10.5% higher accuracy (recall and accuracy with GT is low because non-speaking character regions are not associated with text regions.) This suggests the higher precision in

|          | iter | ctx | cand | prob | Speaker pred. |
|----------|------|-----|------|------|---------------|
| LLM only | 0    |     |      |      | 38.9          |
|          | 0    | ✓   |      |      | 43.6          |
| Multimodal | 1  | ✓   | ✓    |      | 49.1          |
|          | 1    | ✓   | ✓    | ✓    | **51.1**      |

**Table 4: Speaker prediction accuracy with different prompts.**

|                    | Relation pred. | Speaker pred. |
|--------------------|----------------|---------------|
| Distance           | 71.9           | 47.7          |
| SGG (Li et al. 2023) | 78.1         | 50.3          |
| SGG w/ rescoring   | **79.8**       | **51.1**      |

**Table 5: Relationship prediction and speaker prediction accuracy. SGG w/ rescoring is the result after the first iteration.**

pseudo labels affects the character identification performance a lot. In our current method, the combination of the noises of speaker and relationship prediction decreases the precision. Getting reliable pseudo labels is posed as future work.

***LLM Prompts.*** Table 4 shows the results for different LLM prompts in the speaker prediction. The three options in the table correspond to (1) *ctx*: context information about characters and stories, (2) *cand*: speaker candidates, and (3) *prob*: the probability of candidates. Their respective improvements in accuracy are 4.7%, 5.5%, and 2.0%. Each option introduces information not present in the dialogues of each chunk: context provides whole-story information, while the speaker candidates and their probabilities are obtained from the images. These options offer complementary information to each other. The combination of all three options, as shown in Figure 3, performed best.

***Relationship Rescoring.*** Table 5 shows the effect of the proposed relationship rescoring on relationship prediction and speaker prediction. We evaluated the relationship prediction in the manner described in the previous work [8]: for each text region, the corresponding character region with the highest relationship score was selected. Then, the accuracy was calculated by dividing the number of correctly matched regions by the total number of text regions. Our approach with rescoring achieved an accuracy 79.8%, which is better than previous methods. Moreover, it increased speaker prediction accuracy. In contrast to the previous work that doesn't consider character labels, our method can predict the relationships better by predicting character labels of both images and texts.

### 4.4 Zero-shot Evaluation

We evaluated our method in entirely zero-shot settings using only input images. The bounding boxes of the character and the text regions were detected by using the object detection with Faster R-CNN [16], and dialogues were extracted by using OCR in the manner described in [5]. We extracted a list of character names directly from the dialogues using GPT-4. These names then served

|           | iter | Speaker pred. | Character id. |
|-----------|------|---------------|---------------|
| LLM only  | 0    | 34.1          | -             |
| Multimodal | 1   | 37.7          | **35.6**      |
|           | 2    | **38.7**      | 35.0          |
| Upper bound | -  | 67.3          | 63.9          |

**Table 6: Accuracy in an entirely zero-shot setting.**

as target labels for speaker prediction and character identification tasks.

Table 6 shows the experimental results under entirely zero-shot settings. In assessing the experimental results, we manually created mappings between the extracted names and the true names. Each predicted region is counted as correct if it was detected with and IoU >0.5 and was correctly labeled. Our method achieved 38.7% and 35.0% in accuracy for speaker prediction and character identification tasks, respectively. This is lower than the results shown in Table 1 due to errors in object detection, OCR, and character name extraction. Particularly, the character name extraction significantly affects performance because characters whose names are not extracted cannot be predicted. To show the effect of character name extraction on the performance, we compute the *upper bound* in Table 6, which is the accuracy under ideal conditions, where all labels of extracted names are perfectly predicted. The recall for extracted names was 62.7%, thereby limiting the upper bounds to 67.3% for speaker prediction and 63.9% for character identification. It should be noted that there are several titles in which character names are rarely mentioned in dialogues. In that case, it is difficult even for humans to predict the names, which poses the challenges in entirely zero-shot settings. On the other hand, we achieve promising results in titles where character names appear frequently in dialogues and the relationships between characters and texts are clear. Figure 1 shows the results of our approach in entirely zero-shot settings from one specific book, which demonstrate promising first step of zero-shot character identification and speaker prediction of unseen comics.

## 5 CONCLUSION

We are the first to introduce zero-shot character identification and speaker prediction in comics. To solve this unexplored and challenging task, our proposed framework utilizes multimodal integration and high-level context understanding. Since story understanding and multimodal integration are key in comics analysis, our promising results can be a key milestone that opens new directions in comic analysis. In addition, our method can be applied to any comics immediately without the need to train on specific comics, bridging the gap to real-world application. Moreover, our framework can be easily adapted to different domains because each module of our framework is modular and reusable (e.g., learning person classifiers on video data). Since the potential of multimodal speaker prediction extends beyond comics, ranging from movies to online conferences, we hope that our work will impact other fields as well.

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
