# OpenReview forum: "Zero-Shot Character Identification and Speaker Prediction in Comics via Iterative Multimodal Fusion"
_acmmm.org/ACMMM/2024/Conference — MM2024 Oral_

### Official Review · Reviewer_ENLu · 2024-05-13

**Rating:** 4
**Confidence:** 3

**Summary:**

This paper aims to solve the problem of identifying the character name of each person and text box that appears in comic images. The method proposed in this paper includes 2 parts: GPT-4 for speaker prediction, which predicts the speaker given the dialog text, and ResNet50 for character prediction, which predicts the character name given an image region. These two modules are applied iteratively to refine each other. Experiments are conducted on the Manga 109 dataset.

**Strengths:**

1. This paper addresses a new task that has not yet been solved.
2. The authors conducted many ablation studies, which makes it clear how the proposed method works.

**Limitations:**

My biggest concern is the proposed method only works when C (character boxes), D (text boxes), and N (character name list) are manually annotated. As shown in Section 4.4, when this information is not given, the accuracy of this method drastically drops to less than 40%, which means it is hardly applicable in real-world scenarios. Also, the findings of the previous ablation studies may be different in real-world scenarios when those manually annotated information is not given.

I think that it will be better for this study to focus on the “zero-shot evaluation” setting described in Section 4.4, such as how to robustly identify the speakers and characters when N does not exist or is incomplete, and prevent the error accumulation when C and D are not accurate. However, considering that this is the first work for this task, this shortcoming is not unacceptable.

**Suitability:**

3

---

### Official Review · Reviewer_DsXY · 2024-05-23

**Rating:** 3
**Confidence:** 3

**Summary:**

This paper proposes a zero-shot method that is an iterative multimodal fusion framework for character identification and speaker prediction in comics. This approach integrates visual and textual information without requiring any prior annotations or training data. The proposed method employs large language models for text understanding and combines their predictions with image-based classifiers to iteratively refine the identification and prediction tasks.

**Strengths:**

1. This paper discusses an interesting topic and introduces the tasks of zero-shot character identification and speaker prediction. This topic is highly suitable for the MM community.
2. The ability of the proposed method to operate without training data or annotations makes it very practical for real-world applications, where annotating every new comic series is impractical.

**Limitations:**

1. The layout of dialogues in comics varies widely, and different types of text may appear, such as narration or words that the character thinks. Can the proposed method distinguish these complex cases? How do such cases impact the results of speaker prediction?
2. The experimental results indicate that the accuracy of speaker prediction is limited. This makes it challenging for the application of this task.
3. The proposed method seems somewhat more engineering-oriented for the proposed new task.

**Suitability:**

3

---

### Official Review · Reviewer_wws5 · 2024-05-29

**Rating:** 5
**Confidence:** 3

**Summary:**

This paper proposes a zero-shot approach, allowing machines to perform these tasks in new, unseen comics without the need for prior annotations. The framework utilizes an iterative multimodal fusion model to refine predictions through alternate updates of text and image data, leveraging recent advancements in large language models for understanding and reasoning in comic content analysis. The paper aims to establish a robust baseline for character identification and speaker prediction tasks in comics, offering a practical solution applicable to real-world scenarios without the requirement of training data or annotations.

**Strengths:**

This paper integrates character identification and speaker prediction tasks in comics for the first time, and it tackles these tasks as zero-shot learning problems, making the approach directly applicable to real-world scenarios without specific training data. This is the first study to use both text and image information for character identification and speaker prediction, which are unexplored even outside zero-shot settings.

The paper presents an innovative iterative multimodal fusion framework for zero-shot character identification and speaker prediction in comics, which integrates large language models (LLMs) with image-based classifiers to improve predictions through alternate updates. Experiment results highlight the method's capability to handle zero-shot settings and its potential for practical application in comics analysis.

**Limitations:**

Though this work is the first to integrate both character identification and speaker prediction tasks in comics, it lacks sufficient baseline models. It would be beneficial to compare the proposed framework with other large multimodal models (LMMs) such as LLaVA and zero-shot models for speaker prediction.

**Suitability:**

3

---

### Meta-Review · Area_Chair_R3EF · 2024-07-01

**Recommendation:** Accept (Oral)
**Confidence:** 3

**Metareview:**

This paper proposes a multimodal fusion framework for zero-shot character identification and speaker prediction in comics. The task is new, and the proposed approach is interesting. All four reviewers are positive.